# Inverting polar domains via electrical pulsing in metallic germanium telluride

Pavan Nukala[1,*], Mingliang Ren[1,*], Rahul Agarwal[1], Jacob Berger[1], Gerui Liu[1], A.T. Charlie Johnson[1,2] & Ritesh Agarwal[1]

Germanium telluride (GeTe) is both polar and metallic, an unusual combination of properties in any material system. The large concentration of free-carriers in GeTe precludes the coupling of external electric field with internal polarization, rendering it ineffective for conventional ferroelectric applications and polarization switching. Here we investigate alternate ways of coupling the polar domains in GeTe to external electrical stimuli through optical second harmonic generation polarimetry and *in situ* TEM electrical testing on single-crystalline GeTe nanowires. We show that anti-phase boundaries, created from current pulses (heat shocks), invert the polarization of selective domains resulting in reorganization of certain 71° domain boundaries into 109° boundaries. These boundaries subsequently interact and evolve with the partial dislocations, which migrate from domain to domain with the carrier-wind force (electrical current). This work suggests that current pulses and carrier-wind force could be external stimuli for domain engineering in ferroelectrics with significant current leakage.

[1] Department of Materials Science and Engineering, University of Pennsylvania, Philadelphia, Pennsylvania 19104, USA. [2] Department of Physics and Astronomy, University of Pennsylvania, Philadelphia, Pennsylvania 19104, USA. * These authors contributed equally to this work. Correspondence and requests for materials should be addressed to R.A. (email: riteshag@seas.upenn.edu).

Germanium telluride (GeTe), a IV–VI compound, exhibits several unique combinations of properties in the crystalline phase[1–5]. GeTe crystallizes in a rhombohedral structure ($R3m$), characterized by the presence of bonding hierarchy (short and long Ge–Te bonds), which results in a net polarization along the [111] direction, and thus is a polar material[3,6,7]. GeTe also has a large concentration of Ge vacancies ($\sim 10^{19}$–$10^{21}$ cm$^{-3}$) which degenerately dope it with holes, giving rise to p-type metallicity[4,5]. This co-existence of metallicity with the presence of polar axis is very rare in any material system[8,9] and is responsible for exotic properties such as Rashba effect in GeTe[10] and multiferroicity in magnetically doped GeTe systems such as Ge$_x$Mn$_{1-x}$Te (ref. 11). There have been very limited number of studies that attempted to explore and engineer its polar properties[6,7,12–14]. The large carrier density in GeTe screens external field from coupling with internal polarization making it difficult to address and manipulate the polar domains via conventional techniques used for typical ferroelectric systems[15,16]. These issues underline the importance of first developing new techniques to characterize polar domains, and investigating ways to couple these domains to external stimuli. One of the possible external electrical stimulus is electrical current itself. It is important to note that in conventional ferroelectrics, free-carriers annihilate spontaneous polarization[17]. Their co-existence in GeTe, motivates us to seek to answer the question, whether free-carriers (current and also heat) can be used to manipulate the domain polarizations.

Here, we developed a variant of the optical second harmonic generation (SHG) polarimetry[18–20], which is sensitive to material structure in crystals with no inversion symmetry[21–26] to quantify different domain fractions in single-crystalline GeTe. In conjunction with *in situ* transmission electron microscope (TEM) characterization of the GeTe devices[27–30], we studied the interaction of these domains and domain boundaries with extended defects such as anti-phase boundaries (APBs) and associated partial dislocations created by heat shocks from current pulses. Furthermore, we show that neither static fields nor steady currents can couple to the domain polarizations, emphasizing that the only possible external electrical stimulus that can do so are the electrical pulses.

## Results

**Nanowire synthesis and structural characterization.** Single-crystalline GeTe nanowires were synthesized via the vapour–liquid–solid process[31,32] (see Methods) in the $\langle 1\bar{1}0 \rangle$ direction, terminated by two parallel sets of {111} facets along the cross-section (Fig. 1a). Nanowire devices were fabricated on a 50 nm SiN$_x$ membrane based device platform compatible with both TEM and SHG polarimetry experiments[27] (see Methods, Fig. 1a, inset). For the SHG polarimetry experiments which require the usage of high incident laser powers (average power, few GW cm$^{-2}$)[33], a heat sink for the devices was created by a conformal deposition of 30 nm Al$_2$O$_3$ (insulating oxide), followed by 100 nm of Ag. The {111} termination ensured that the incident beam (both laser and electron) was always shone along the $\langle 111 \rangle$ direction (zone axis; see Supplementary Fig. 1 for SHG experimental set-up).

The stable structure of GeTe can be understood as a rock salt structure rhombohedrally distorted along one of its body diagonals, [111], with a net polarization along that direction[4,5]. The four body diagonals along which the distortion of the rock salt structure can occur, combined with the possibility of two polarization directions along each body diagonal (with an angle of 180° between them), can give rise to eight different polar domains (four ferroelastic domains), which in general coexist in as-grown samples[34]. These eight domains will be addressed from here on as

$\alpha(+,-)$, $\beta(+,-)$, $\gamma(+,-)$ and $\delta(+,-)$, each of them identifiable with the direction of the polarization vector. The Greek letters $\alpha$, $\beta$, $\gamma$ and $\delta$ refer to the ferroelastic domains and $+$ or $-$ refers to their polarization direction. We label domains as positive, if their polarization vectors are pointing out of the substrate, and as negative if they point into the substrate. $\alpha(+,-)$ are a pair of inversion domains with the polarization vector ($\mathbf{P}_{\alpha(+,-)}$) parallel to the zone axis (optical viewing direction). $\beta(+,-)$ are the inversion pair of domains whose polarization directions $\mathbf{P}_{\beta(+,-)}$ are perpendicular to the set of terminating {111} facets not parallel to the substrate. Domains $\gamma(+,-)$ and $\delta(+,-)$ have polarization directions $\mathbf{P}_{\gamma(+,-)}$, $\mathbf{P}_{\delta(+,-)}$ in the other two $\langle 111 \rangle$ directions respectively, not perpendicular to any of the nanowire facet (Fig. 1a). Within every polar domain, there can exist two different cationic (anionic) stacking sequences (ABCABC… or ACBACB…) along the polarization direction, resulting in two stacking domains (ferroelastic), 1 and 2, related by 180° rotation along the polar axis (Supplementary Fig. 2). Hence in totality there are sixteen domains in GeTe, $\alpha(+,-/1,2)$, $\beta(+,-/1,2)$, $\gamma(+,-/1,2)$, $\delta(+,-/1,2)$, and obtaining information from individual domains is challenging.

Domain walls in GeTe can be classified as 71° ({001} habit planes) or 109° ({1$\bar{1}$0} habit planes) boundaries based on the angle between the polarization vectors of the constituting domains. We simulated the selected area electron diffraction (SAED) patterns of these domain walls viewed along $\langle 111 \rangle$ zone axis to understand their differential signatures (see Methods). Upon superimposing the SAED of two domains forming the {001} and {1$\bar{1}$0} boundaries, we note that the diffraction spots split in $<11\bar{2}>$ and $\langle 1\bar{1}0 \rangle$ directions respectively (Fig. 1b,c). Conversely, the nature of the spot splitting can be used to identify the type of the domain wall. As-synthesized GeTe nanowires contain {001} boundaries (parallel to the growth direction) between $\gamma(+/-)$ and $\delta(-/+)$ domains as evidenced by the spot splitting along $\langle 11\bar{2} \rangle$ direction (Fig. 1d,e). It must be noted that while diffraction-contrast TEM provides structural information on domains and domain walls illuminated in a particular zone, a suitable SHG polarimetry technique gives quantitative information on all the domain fractions[18,20].

**Second harmonic generation polarimetry on GeTe.** Developing SHG polarimetry on GeTe is challenging owing to the lack of standard samples (single-domain) and the non-linear material constants ($R3m$ $\chi^{(2)}$ tensor, Supplementary Note 1). Single-crystalline GeTe samples, though not standard, are the best option, although complications arise owing to the measured SHG signal being an interference of signals from sixteen domains. We designed polarimetry experiments[18] on single-crystalline nanowires where the fundamental light (1,020 nm, spot size: 2 μm) was shone along the $\mathbf{P}_{\alpha(+)}$ direction (Fig. 1a, inset) and we varied its linear polarization direction from $-180°$ to $180°$ with respect to the long-axis of the nanowire. SHG signal (510 nm) was collected in the reflection mode at three particular polarizations, that is, 0° ($\mathbf{x}_0$), 60° ($\mathbf{x}_{60}$) and 120° ($\mathbf{x}_{120}$) with respect to the long-axis of the nanowire, corresponding to $\langle 110 \rangle$ directions in every domain (Fig. 1d,e) (see Methods, Supplementary Note 1, Supplementary Fig. 1 for details). The collection of SHG signal along three different polarizations is unique to our polarimetry experiments, and enables separation of $\beta$, $\gamma$ and $\delta$ domains whose axis of three-fold symmetry is not the zone axis (optical viewing axis). These intensities were fit to the expression:

$$I(2\omega, \theta) = [A_\theta \cos^2(\phi - \theta) + B_\theta \sin^2(\phi - \theta) + C_\theta \sin 2(\phi - \theta)]^2$$

(1)

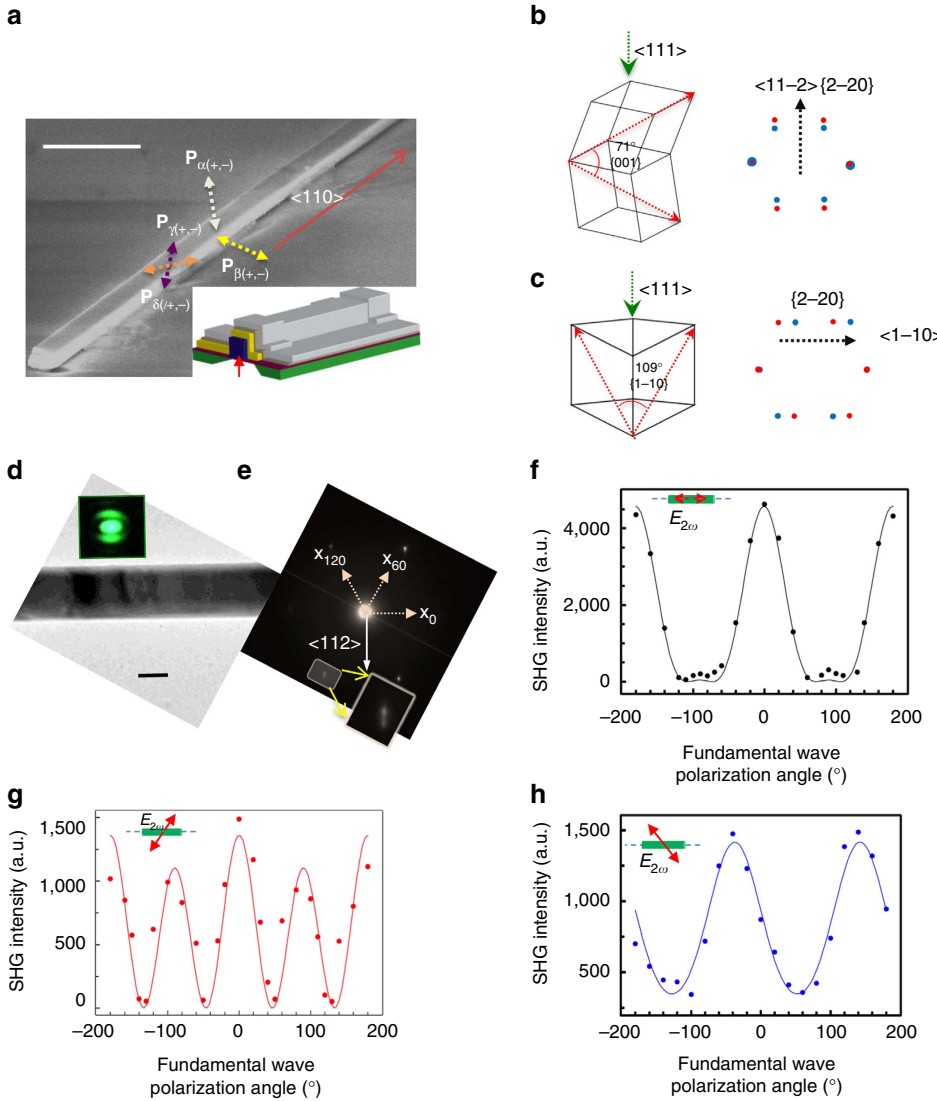

**Figure 1 | Quantifying and characterizing domains in single-crystal GeTe. (a)** Scanning electron microscope image of a representative GeTe nanowire showing the {111} terminating facets parallel to the growth direction ⟨1̄10⟩. The polarizations of ferroelastic domains α, β, γ, δ domains are indicated. $\mathbf{P}_{\alpha(+)}$ and $\mathbf{P}_{\alpha(-)}$ are the polarizations of inversion pair of domains α(+) and α(−) respectively, perpendicular to the {111} facet on the substrate. $\mathbf{P}_{\beta(+)}$ and $\mathbf{P}_{\beta(-)}$ are the polarizations of variants of inversion domains, β(+) and β(−) perpendicular to the other {111} facet. Similarly, $\mathbf{P}_{\gamma(+,-)}$ and $\mathbf{P}_{\delta(+,-)}$ are the polarizations of variants of γ and δ respectively along the other two ⟨111⟩ directions, not perpendicular to any of the nanowire facet. All positive polarization vectors point away from the substrate, whereas negative polarization vectors point into the substrate. Scale bar; 2 μm. (inset) Schematic of the device used for SHG polarimetry. Device cross-section consists of the following layers: from bottom to the top: Si substrate (green), SiN$_x$ membrane (maroon), GeTe nanowire (violet), electrodes (yellow), Al$_x$O$_y$ (gold) and Ag heat-sink (silver). Fundamental light is shone as indicated in the red arrow. **(b,c)** Schematics showing a {001} **(b)** and {1̄10} **(c)** domain walls between two domains (say γ(+) and δ(−)), and selected area electron diffraction (SAED) pattern obtained by superimposing the constituting domains (red and green spots) showing spot splitting along ⟨112̄⟩ (black arrow) in **c** and ⟨1̄10⟩ in **c**. **(d)** Bright-field TEM image of an as-grown nanowire, and its strong SHG signal (inset). Scale bar; 200 nm. **(e)** SAED pattern of the representative nanowire shown in **d**, showing spot splitting along ⟨112̄⟩ (inset), an indication of presence of {001} domain walls between γ and δ. **(f–h)** SHG polarimetry data collected by varying fundamental polarization, and fixing the SHG polarizer at **(f)** θ = 0°, **(g)** θ = 60°, and **(h)** θ = 120° with respect to the long axis of the nanowire (which are the different ⟨1̄10⟩ directions x$_0$, x$_{60}$ and x$_{120}$ respectively in **d**). Fits to the data are shown in solid lines. Table 1 shows the material constants and domain fractions obtained by fitting the data in **f–h** to equation (1), and solving them from the known fitting parameters.

where θ = 0, 60 or 120° is the SHG polarization angle, and $\phi$ is the polarization of the fundamental wave (for derivation, see Supplementary Note 1, Supplementary Fig. 3). Material constants and domain fractions were obtained from the fitting parameters ($A_\theta$, $B_\theta$, $C_\theta$) (Supplementary Note 1). For the virgin nanowire whose polarimetry data is shown in Fig. 1f–h and Table 1, the material constants were obtained as $d_{31}/d_{15} = 1.26$ and $d_{15}/d_{33} = -0.72$, where $d_{ij}$ refer to elements of the third order tensor (in Voigt notation) relating SHG response to the

incident field (Supplementary Note 1). The ratios of volume fractions of domains following the same stacking sequence (either 1 or 2) were calculated as shown in Table 1, with the understanding that $V_{d(i)} = V_{d(+,i)} - V_{d(-,i)}$, where 'd' labels the ferroelastic domain α, β, γ or δ, and i = 1 or 2 labels the type of stacking sequence (Supplementary Note 1). On 15 different nanowire devices of large diameters (>500 nm to avoid anisotropy in in-coupling and out-coupling of light)[18,33,35]; we estimated the median value of the material constants

$d_{31}/d_{15}$, and $d_{15}/d_{33}$ to be 1.05 and $-0.53$, respectively (Supplementary Fig. 4).

**Domain dynamics with static electric field.** Engineering domains electrically in polar metals such as GeTe, requires finding suitable external electrical stimuli that can couple to spontaneous polarization. To understand if static fields or steady currents have any effect at all on the internal polarizations, we performed *in situ* TEM by recording structural changes in GeTe devices (Supplementary Movie 1, and Supplementary Fig. 5) while sweeping the d.c. voltage from 0 V until device failure. The structural changes for a nanowire device with a resistance of 1.85 kΩ (Fig. 2a) was recorded in the bright-field (BF) imaging mode (Supplementary Movie 1, Fig. 2b), and there were no observable changes in the diffraction contrast up to the point where the nanowire melts and sublimates ($I = 0.75$ mA) leaving behind $Al_xO_y$ shell (Fig. 2c,d), that was conformally deposited for stable switching[29]. We recorded movies on two other nanowires one in diffraction mode (snapshots in Supplementary Fig. 5a), and another on a thinner nanowire ($\sim 120$ nm) in a bright-field mode (snapshots in Supplementary Fig. 5b) and observed no discernable structural changes until device failure. These findings clearly reveal that the electric field and associated d.c. current (and Joule heating) cannot couple with the internal domain polarizations.

**Domain inversion and dynamics with current pulses.** GeTe, a phase-change material, can exist in either a stable crystalline phase or a metastable amorphous phase and can be switched back and forth between the two phases, giving rise to a memory functionality[36]. The reversible transformation between these phases happens via the application of electrical pulses, and a defect-based pathway for crystal-amorphous phase transformation has been recently discovered in phase-change materials such as $Ge_2Sb_2Te_5$ and GeTe[27–30]. In this pathway, the application of electrical pulses (heat shocks) on GeTe accumulates Ge vacancies in a {111} plane, which beyond a certain size condense causing a local collapse of adjacent Te planes. This results in the creation of an APB, a 2D translational defect, which is surrounded by partial dislocations (with Burgers vector, $\mathbf{b} = \frac{1}{4}\langle 1\bar{1}0 \rangle$)[29]. These partials migrate along suitable planes in the direction of electrical current due to the transfer of momentum from hole carriers (the hole-wind force), until they encounter a region of local inhomogeneity where all these defects accumulate forming an immobile defect template. Addition of more defects to this template eventually results in the collapse of long-range order in a local region, resulting in the nucleation of an amorphous phase[29]. It is hence possible that extended defects such as APBs and partial dislocations, which can be created and controlled by heat shocks and carrier-wind force, can interact with the polar domains and manipulate them, thus rendering electrical pulses as an external parameter that can engineer domains.

To verify whether heat shocks through electrical pulses can couple with the polar domains, we performed *in situ* TEM analyses on GeTe nanowire devices in the dark-field mode (reciprocal lattice vector, $\mathbf{g} = \langle 02\bar{2} \rangle$ or $\mathbf{x}_{120}$), where the domain boundary evolution was tracked while simultaneously applying a train of voltage pulses (200 ns pulse width, 1 s pulse separation) of increasing amplitude until amorphization (Supplementary Movie 2, snapshots Fig. 3a–j). The SAED of the virgin state of the device (Fig. 3a,b) showed split spots along $\langle 11\bar{2} \rangle$ direction, perpendicular to the growth axis, consistent with the existence of {001} domain boundaries between $\gamma(+/-)$ and $\delta(-/+)$. The {001} domain boundary contrast does not explicitly appear in the dark-field images, owing to them not being in the zone defined by $\langle 111 \rangle$ zone axis. However, the line defects along the nanowire, pre-induced by focus ion beam (FIB), act as tracers to the {001} domain wall dynamics (Fig. 3c,d). While not much change in

| Table 1 \| Domain volume fractions on a representative nanowire. | |
|---|---|
| $d_{33}/d_{15}$ | 1.26 |
| $d_{15}/d_{31}$ | $-0.72$ |
| $V(\beta_1):V(\gamma_1):V(\delta_1)$ | $1:0.67:-2.08$ |
| $V(\beta_2):V(\gamma_2):V(\delta_2)$ | $1:0.29:-1.61$ |

SHG, second harmonic generation.
Values of $d_{33}/d_{15}$ and $d_{15}/d_{31}$, and the volume fraction of various domains on a nanowire whose SHG polarimetry data is shown in Fig. 1f–h. These were obtained by fitting the polarimetry data to equation (1), via a procedure described in Supplementary Note 1. Note that the volume fractions are normalized with respect to that of β domain.

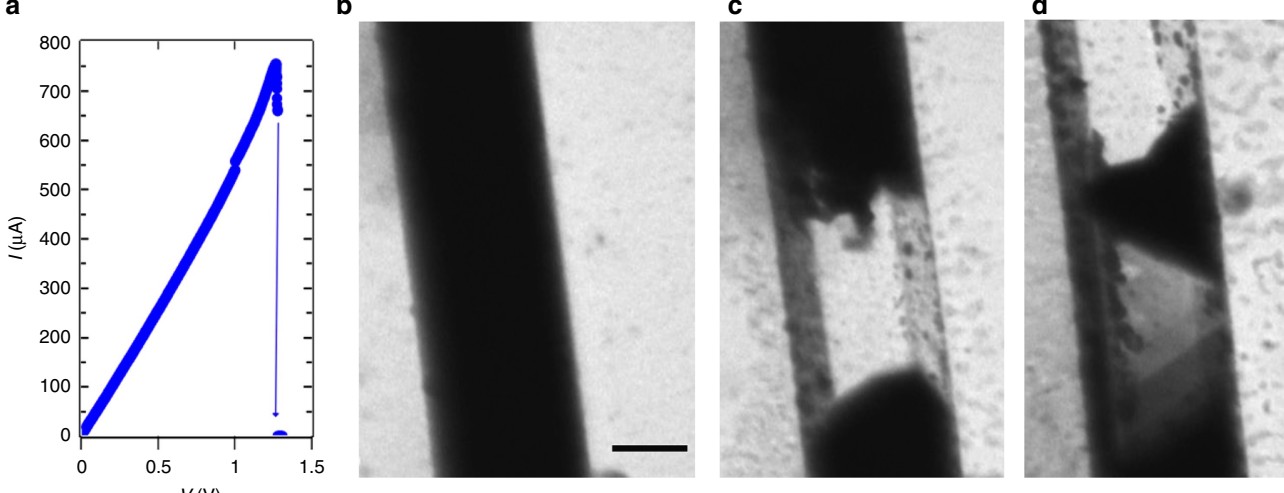

**Figure 2 | *In situ* TEM voltage-controlled I–V sweep on a GeTe nanowire device.** (**a**) I–V sweep on a device with 1.85 kΩ resistance, from 0 to 1.34 V when the device fails. (**b,c**) Snapshots from bright-field TEM Movie 1, recorded on the device during the d.c. voltage sweep. (**b**) From 0 to 1.34 V no changes in structure are observed. (**c**) At 1.34 V, 0.75 mA current passes through the device, melting and sublimating it. So, neither the field nor the high d.c. current and associated Joule heating, couple to the internal domain polarizations. (**d**) Liquid GeTe flows inside confined to the conformal $Al_xO_y$ coating altering the contrast from (**c**). Scale bar; 200 nm.

contrast was observed until a pulse amplitude of 2.5 V, Supplementary movie 2 shows reorientation of these pre-induced defects and hence the {001} domain walls from 2.6 to 2.76 V pulse voltage range. At 2.76 V, we observed the nucleation of another domain wall contrast perpendicular to the growth axis of the nanowire (Fig. 3e), which subsequently grew into a periodic pattern (Fig. 3f). SAED performed in this periodic contrast region after the application of 2.86 V pulse showed a spot splitting in the $\langle 1\bar{1}0 \rangle$ direction (growth axis), revealing that the observed periodic contrast is from $\gamma(+/-)$ and $\delta(+/-)$ domains separated by $\{1\bar{1}0\}$ boundaries (Fig. 3g). Hence the {001} or 71° boundaries between $\gamma$ and $\delta$ domains reorganize into perpendicular $\{1\bar{1}0\}$ boundaries upon the application of electrical pulses beyond a certain voltage, demonstrating that heat shocks can manipulate the polar domains. Following this, we observed partial dislocation migration leading to amorphization across the nanowire cross-section at the notch at 3.16 V (ref. 29; Fig. 3h,i). The partial dislocations near the amorphized region (Fig. 3i) could not be identified owing to the strong contrast of the amorphous phase (see Supplementary Note 2 and Supplementary Fig. 6 for identification of various defects in dark-field microscopy videos). However, this contrast became very clear upon the application of 3.2 V pulse, which crystallized the metastable amorphous region via heating it beyond crystallization temperature (Fig. 3j). When the same device was reprogrammed again towards the amorphous phase, we could clearly see the migration of partial dislocations (>1 V), along the direction of hole-wind, from domain to domain resulting in altering and eventually blurring the $\{1\bar{1}0\}$ domain walls (>2.7 V, Supplementary Movie 3, Fig. 4a–d). When

these partials accumulate near the notch region creating an immobile defect template (leading to amorphization), the domain structure locally was severely affected, as suggested by the diffuse diffraction spots in SAED[27–29,37] (Fig. 4e, HRTEM images in Supplementary Fig. 6d). These results illustrate that APBs and partial dislocations created by heat shocks (see HRTEM evidence in Supplementary Fig. 7) can be used to address and manipulate polar domains in polar metals such as GeTe. It is important to note that, while the creation of notch localizes the region of observation and creates defects that can track {100} domain boundaries, similar defect evolution occurs in devices without the notch, and the morphological inhomogeneity for defect accumulation is provided by cathode- nanowire interface[27] or domain boundaries.

To obtain more quantitative information on the changes in domain fractions in GeTe upon electrical pulsing and understand if such changes correspond to the domain wall reorganization observed via in situ TEM, we performed SHG polarimetry on nanowire devices at different positions in the virgin state and also after applying electrical pulses until amorphization (Fig. 5a–e). For the nanowire device shown in Fig. 5a, polarimetry data was obtained from positions 1 and 2. Position 1 is closer to the cathode, and is in a state of compressive stress which favors reduced defect mobility, and hence is a very likely location for defect template formation and amorphization upon programming[27]. Position 2, however, is at the center of the device where domain wall reorganization occurs but not defect accumulation (Fig. 3). The virgin state showed strong SHG signal (Fig. 5b) and similar polarimetry behavior at both positions 1

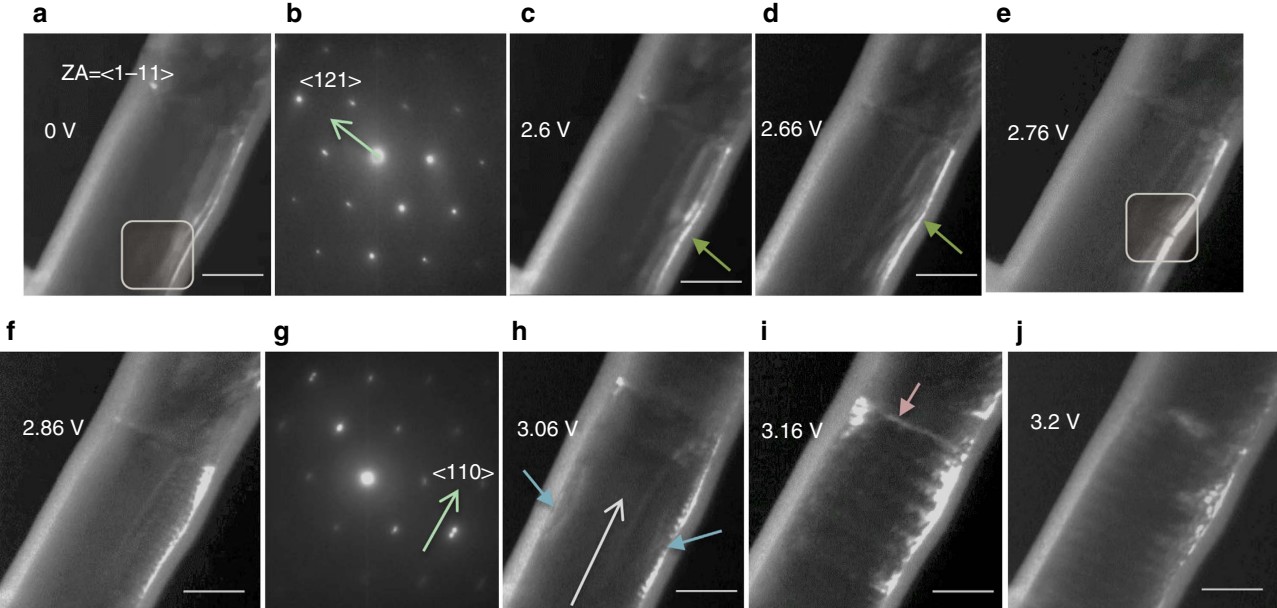

**Figure 3 | In situ TEM analyses of domains in GeTe nanowire devices with electrical pulses.** (a–g) Snapshots of first crystal-amorphous transformation event from Supplementary Movie 2, recorded in dark-field imaging mode, with the reciprocal lattice vector selected for imaging, **g** being $\langle 02\bar{2} \rangle$, at 120° to the growth axis ($x_{120}$). (a) Dark-field (DF) TEM image of a nanowire device in virgin state with **g** indicated in yellow in the SAED (b). Defects induced during sculpting a notch via focused ion-beam, can be seen inside the rectangle. (b) SAED pattern from a region in the nanowire (indicated) shows spot splitting in the $\langle 11\bar{2} \rangle$, suggesting the existence of {001} boundaries between variants of $\gamma$ and $\delta$ domains in the virgin state. (c,d) Dark-field TEM snapshots of the device showing the evolution of FIB induced defects, which trace the trajectory of {001} domain walls with increasing voltage pulse amplitude. The pre-induced defects trace the {001} domain boundaries which are otherwise difficult to see in this zone are pointed by closed yellow arrows. (e) Nucleation of $\{1\bar{1}0\}$ boundaries (boxed) and (f) growth of $\{1\bar{1}0\}$boundaries. Spot-splitting in $\langle 1\bar{1}0 \rangle$ in SAED (g) evidence of transformation of {001} boundaries in **a** to $\{1\bar{1}0\}$ boundaries in **e**,**f**. (i,j) Partial dislocation migration along the nanowire long axis leading to amorphization at 3.16 V pulse amplitude. In **h** partial dislocation are indicated with blue closed arrows, while their migration direction is shown with open white arrow. Amorphous region is indicated in i. While the contrast from partial dislocations is not so clear with respect to the amorphous phase in **i**, it becomes very clear once the device recrystallized upon the application of a 3.2 V pulse in (**j**, closed blue arrows). Scale bars; 100 nm.

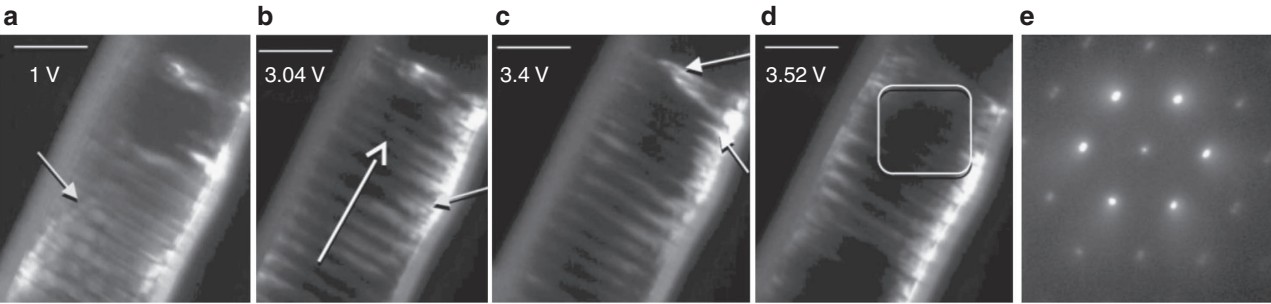

**Figure 4 | Interaction of hole-wind force with domains.** (**a–d**) Snapshots of another crystal-amorphous transformation event on the same device from Supplementary Movie 3, recorded in the same dark-field conditions as Supplementary Movie 2. (**a**) Beginning of partial dislocation migration in the direction of hole- wind force between various polar domains. Partials are indicated by sky-blue arrows. (**b**) Migration of partials at higher voltages causing $\{1\bar{1}0\}$ domain wall alteration and blurring. White open arrows point at the direction of partial dislocation migration. (**c**) Accumulation of partials creating an intersecting template of immobile defects (pointed by closed arrows). (**d**) Amorphization event at a local region in this template. (**e**) SAED from the non-amorphous region in the template shows diffuse spots, a signature or heavy disorder. Scale bars; 100 nm.

and 2 (Fig. 5c–e). Upon programming to amorphization, position 1 produced a very weak SHG signal (Fig. 5b), consistent with the existence of intersecting template of defects and ill-defined domains. However, position 2 showed polarimetry response different from virgin state. A full computation, as described in Supplementary Note 1, revealed a significant inversion of the variants of $\delta$ domains upon the application of electrical pulses, and a slight change in variants of the $\gamma$ domain after programming (Table 2, see Supplementary Fig. 8 and Supplementary Table 1 for data on another device).

The following aspects of equation (1) enable us to qualitatively predict the changes in domain fractions from the virgin to programmed state: (i) The domain whose polarization is perpendicular to SHG polarizer, contributes only to the coefficient of $\sin 2(\phi - \theta)$ ($C_\theta$) of the total intensity (that is, variants of $\beta$, $\gamma$ and $\delta$ for $\theta = 0^o$, $60^o$ and $120^o$ respectively), and the rest of the domains contribute to coefficients of all the three terms. (ii) Coefficient of $\sin 2(\phi - \theta)$ or $C_\theta$ produces a periodicity of $\pi/2$ to produce four lobes in polarimetry plots for $\phi$ varying from $-180$ to $180^o$, while coefficients of $\cos^2(\phi - \theta)$ and $\sin^2(\phi - \theta)$ ($A_\theta$ and $B_\theta$) produce two lobes. The change in phase of the fits at $\theta = 60^o$ (Fig. 5d), predominantly dominated by coefficients of $\sin^2(\phi - 60^o)$ and $\cos^2(\phi - 60^o)$ (two lobes) between virgin and programmed states suggests significant changes in domain fractions of either the variants of $\beta$ or $\delta$. Figure 5e shows no significant differences between the two states at $\theta = 120^o$, both fit to a two lobed function, except for a stronger signal in virgin state. This indicates that neither the variants of $\beta$ nor $\gamma$ change significantly upon programming, and in conjunction with the polarimetry plots for $\theta = 60^o$ (Fig. 5d), a significant change in variants of $\delta$ may be predicted. This is also consistent with the $\theta = 0^o$ (Fig. 5c) polarimetry plot, where changes in variants of $\delta$ enhance the contribution of coefficients of $\sin^2(\phi)$ and $\cos^2(\phi)$ (responsible for two lobes) in comparison with the coefficient of $\sin(2\phi)$ (responsible for four lobes) post programming.

## Discussion

Heat shocks from a current pulse cluster Ge vacancies in a $\{111\}$ plane, and beyond a certain cluster size, the Te planes above and below collapse forming an APB, a 2D extended defect[29]. The formation of APB in a particular domain creates two anti-phase domains, one of which subsequently relaxes by altering the sequence of long and short bonds inverting its polarization by $180^o$ (Fig. 5f). Furthermore, upon the nucleation of an APB (say in the $\delta(-/+)$ domain) near a pre-existing $\{001\}$ domain wall

separating the variants of $\gamma(+/-)$ and $\delta(-/+)$ domains (domain boundaries can provide nucleation sites for APBs), $\delta(-)$ completely switches its polarization by $180^o$ to nucleate $\delta(+)$. This polarization inversion, as confirmed from SHG polarimetry, results in the reorganization of the $\{001\}$ boundary into $\{1\bar{1}0\}$ boundary between variants of $\gamma$ and $\delta$ domains (Fig. 5g), consistent with *in situ* TEM observations. Further application of electrical pulses creates more APBs, which results in more $\delta(+)$ and its ferroelastic rearrangement with $\gamma(+)$ (evidenced by the slight change in the domain fraction of $\gamma(+)$ (Table 2), forming a periodic arrangement of these domains in a diffusionless process[38]. It must be realized that the significant inversion of $\delta$ domains observed in Fig. 5c–e and Table 2, arises from APB nucleation in the $\delta$ domain. However, it is equally probable for the first APB nucleation in the $\gamma$ domain, resulting in its inversion. The deterministic part in this domain inversion is that only one of $\gamma$ or $\delta$ inverts its polarization. While APB formation via vacancy accumulation is a diffusion assisted process whose kinetics are expected to non-linearly increase with increasing voltage[39], ferroelastic relaxation that follows should occur very fast, at the speed of sound (1,900 ms$^{-1}$) in GeTe[40]. APBs are also sources of partial dislocations[29,41] (HRTEM data in Supplementary Fig. 7), which migrate with the hole-wind force from domain to domain. Domain wall acts as an impediment to the motion of partial dislocations, thus reducing their mobility. Owing to this, eventually there will be more partial dislocation concentration at each domain wall compared to inside the domain. This disorder causes the already incoherent $\{110\}$ domain walls to further blur out (Fig. 3j–l). The direction of the hole-wind provides some determinism in predicting the domains that invert their polarization and their further interaction with the partial dislocations.

The current study can lead to exploring several intriguing questions in leaky polar materials. For instance, given the crystallographic similarities of the various types of domain walls ($71^o$, $109^o$, $180^o$) with a well-studied rhombohedral ferroelectric BiFeO$_3$ (BFO)[42], certain questions and comparisons about the nature of these domain walls with BFO can make an interesting study. Specifically, are the $180^o$ boundaries mobile or immobile, and are they charged or uncharged? The $180^o$ boundaries contain a plane of Te-Te bonds (Fig. 5g), with head to head polarizations on either side of the domain wall, resulting in some extra bound charge at the domain boundary itself, although likely screened by the presence of a sea of electrons. Furthermore, our movies (Supplementary Movie 3) suggest that it is only the partial dislocations that move easily with hole-wind force and not the domain boundaries themselves. The reason for this may stem

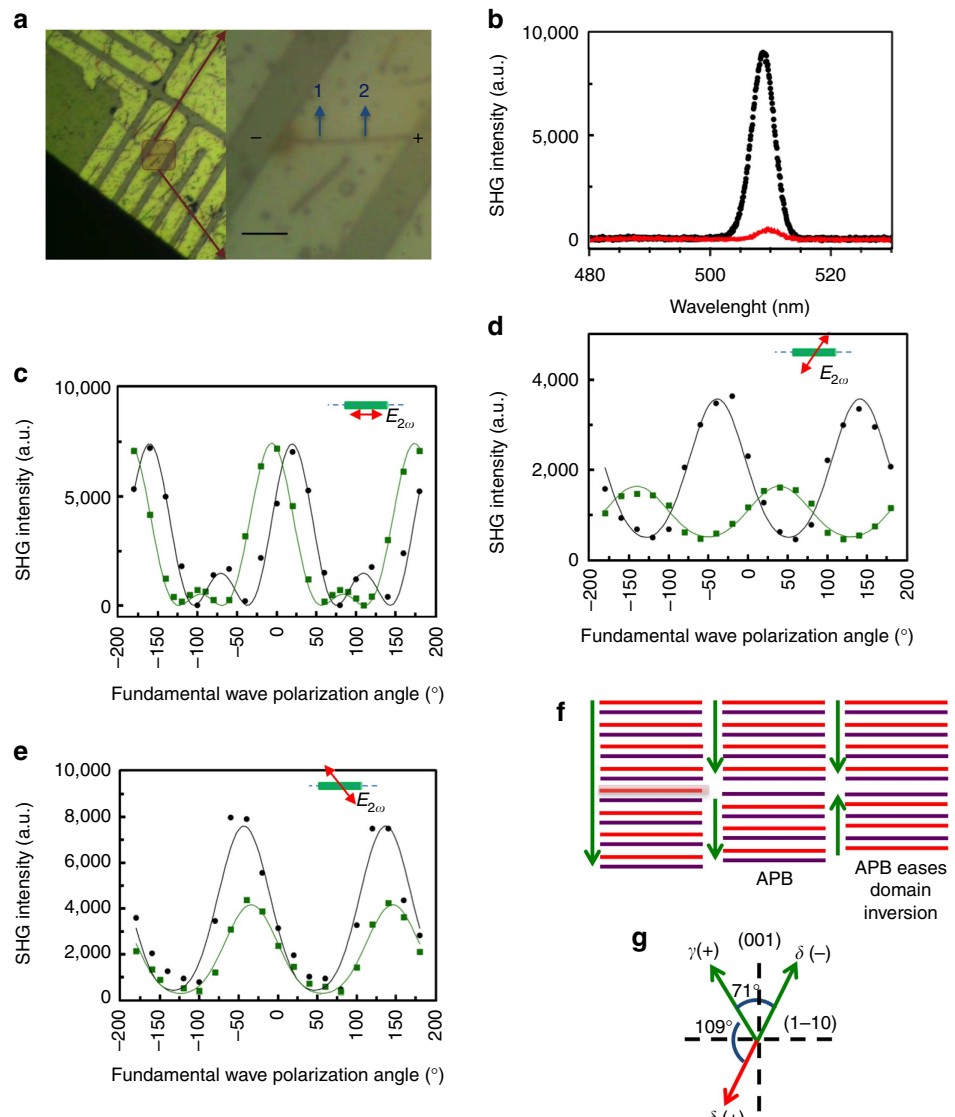

**Figure 5 | Evidence of domain inversion after programming via SHG polarimetry.** (**a**) Optical microscope image of the nanowire device platform, zoomed into nanowire device of interest with anode and cathode contact regions marked. Positions 1 and 2 are the locations on the nanowire device where polarimetry experiments were performed in virgin state, and after the application of electrical pulses to amorphize a local region. Scale bar; 10 μm. (**b**) SHG intensity (without SHG polarizer) at position 1 showing very strong signal from the virgin state (black) in contrast to very weak signal after the application of electrical pulses (red), suggesting the formation of intersecting defect template near position 1 post programming. (**c–e**) SHG polarimetry behavior comparison at position 2 between virgin (black) and programmed (green) states at (c) $\theta = 0^{\circ}$ ($\mathbf{x}_0$), (**d**) $\theta = 60^{\circ}$ ($\mathbf{x}_{60}$), and (**e**) $\theta = 120^{\circ}$ ($\mathbf{x}_{120}$) with respect to the nanowire long-axis, revealing a significant inversion of variants of $\delta$ ( − ) to $\delta$ ( + ), and a slight change in variants of $\gamma$ ( + ) (Table 2). (**f**) Schematic showing how APB formation (collapse of Te plane above and below onto a Ge vacancy plane) can facilitate domain inversion. The red and violet lines represent Ge and Te planes respectively, and the green arrows represent the domain polarization. APB is formed by accumulation of Ge vacancies in a Ge plane, eventually resulting in the collapse of two Te planes (middle panel). Formation of APB creates two anti-phase domains, one of which upon slight relaxation can invert its polarization (right panel), by altering the sequence of long and short bonds between Ge and Te planes. (**g**) Schematic explaining how polarization inversion from $\delta$ ( − ) to $\delta$ ( + ) transforms the nature of the domain boundary between $\gamma$ and $\delta$ from {001} to {1$\bar{1}$0}.

| | Virgin (position 2) | After programming (position 2) |
|---|---|---|
| $d_{33}/d_{15}$ | 1.44 | 1.57 |
| $d_{15}/d_{31}$ | − 0.17 | − 0.35 |
| $V(\beta_1):V(\gamma_1):V(\delta_1)$ | 1:1.60: − 1.23 | 1:1.03:0.92 |
| $V(\beta_2):V(\gamma_2):V(\delta_2)$ | 1:1.49: − 0.34 | 1:1.02:1.05 |

**Table 2 | Domain inversion after programming on a nanowire device.**

Material constants and volume fractions of a GeTe nanowire device shown in Fig. 5a, position 2, before and after electrical pulsing. The polarimetry data on this nanowire is shown in Fig. 5c–e.

from the immobility (or the large energy expense required to move them) of the 180° boundaries (formed from APBs). So, while this study provides evidence that 180° boundaries are probably immobile, and an expectation that they are charged, these hypotheses need to be confirmed with further studies.

The real challenge in these polar metals, however, is to be able to reverse the polarization switching. Once APBs are formed in the $\delta$ domain, its polarization inverts, and transforming the nature of the domain boundary between $\gamma$ and $\delta$ from 71° ($\gamma$( + )/$\delta$( − ) interface, say) to 110° ($\gamma$( + )/$\delta$( + ) interface) type. Thus it seems possible that upon nucleation of another APB in the $\gamma$ domain, domain boundary flips again to 71° type

($\gamma(-)/\delta(+)$ interface), and this could go on forever. However, in our nanowire geometry once the APBs are nucleated on a domain (say $\delta$), the partial dislocation migration[29] becomes the dominating process, rather than creation of APBs in other domains. Generalized stacking fault (GSF) energy calculations in GeTe clearly show that upon the existence of a source for partial dislocations with Burgers vector $\frac{1}{4}\langle 110 \rangle$, the GSF for their migration path in the {111} plane is 217.5 mJ m$^{-2}$, an energy comparable to GSF in Cu (a pure metal with dislocation propagation being the well-known mechanism for plastic deformation) of 175 mJ m$^{-2}$ (ref. 29). Hence reversing the domain boundary flip cannot simply happen by reversing the direction of the hole-wind force. Rather, this requires careful studies where the hole-wind is confined to directions along which partial dislocation motion is harder, and the system prefers to nucleate more APBs instead.

The free-carriers in conventional ferroelectric materials such as BaTiO$_3$ seek to annihilate ferroelectricity[17] by screening the Coulombic long-range interactions[43] which are responsible for spontaneous polarization. In GeTe, however, this spontaneous polarization is provided by electron-phonon interaction[44], which can co-exist with p-type carriers in the valence band. At a fundamental level, our experiments were designed on the premise that, since the free-carriers do not annihilate polarization, perhaps they could be utilized to engineer the domains. Our observations of current pulse and hole-wind assisted extended defect interactions with domain polarizations is a step towards the ultimate goal of their control deterministically and reversibly in GeTe in particular, and other leaky polar materials in general. However, since we utilize heat (or heat shocks) via current pulses to perform work required for domain switching, some inefficiencies and losses are expected, compared to switching a conventional ferroelectric such as BaTiO$_3$ using electric field.

Finally, any commercial applicability for polarization switching in these polar metals arises only if there are well-defined changes in measurable properties associated with switching. While FeRAM applications may seem far-fetched, specifically, in case of GeTe, it is recently reported[10] that a 180° domain switching results in a change in the Fermi level (and hence the work function) of a Te terminating plane by $\sim 200$ meV, potentially making domain switching in thin films of GeTe useful for tunnel junction applications. Furthermore, the combination of polar metallic nature makes dilute magnetic materials such as Mn doped GeTe, where magnetic RKKY interactions arise from free-carriers, multiferroic[11].

In conclusion, we demonstrated that extended defects created from heat shocks such as APBs and partial dislocations interact with polar domains and can be used to manipulate them in polar metals such as GeTe. In particular, by developing the SHG polarimetry technique on GeTe and utilizing it in conjunction with *in situ* TEM, we showed that the formation of an APB inverts polarization of selective domains, and domain walls are altered via interactions with partial dislocations, which can be externally controlled by current pulses. Furthermore, we clearly demonstrate that heat shocks or electrical pulses can couple to internal polarizations and address them in these materials. Leakage issues in ferroelectric materials preclude their efficient functionality, and strategies using chemical doping have been sought to reduce leakage[45,46]. However, this work shows that extended defects that interact with domain polarizations and respond to current pulses provide a mechanism for domain engineering in polar materials with such leakage issues.

## Methods

**GeTe nanowire synthesis.** GeTe nanowires were synthesized using metal catalyst mediated vapour–liquid–solid process. Bulk GeTe powder (99.9%, Alfa Aesar,

$T_m = 724$ °C) was placed at the center of a tube furnace. Silicon (001) substrate evaporated with Au film (15 nm) and annealed at 720 °C for 10 min, was placed on the downstream side of the furnace ($\sim 15$ cm away from the middle). The furnace was ramped up to 720 °C, with 25 sccm of Ar (carrier gas) flow rate and a pressure of 120 torr. The reaction was carried out for an hour, after which the furnace was slowly cooled down to room temperature.

**Device fabrication.** The *in situ* TEM holder was home-built and compatible with JEOL 2010F and 2100 TEMs, and characterization using this holder requires a special nanofabrication procedure of GeTe nanowire devices on an electron transparent SiN$_x$ window[27]. Double side polished Si wafer, with 300 nm SiN$_x$ film deposited on both sides via low-pressure chemical vapour deposition were used as substrates to fabricate *in situ* TEM and SHG polarimetry compatible nanowire devices[27]. A window (300 μm × 300 μm) of 300 nm thick SiN$_x$ membrane was created via dry-etching a larger window of 300 nm thick SiN$_x$ from the backside, followed by KOH based wet etching $\sim 600$ μm Si. Ti (5 nm)/Au (30 nm) electrodes were patterned on the membrane using photolithography. Nanowires were dry transferred onto the membrane, and two terminal devices were fabricated via Pt deposition using FIB technique. The devices were subsequently annealed at 250 °C for an hour, and $\sim 15$ nm Al$_2$O$_3$ film (insulating oxide, κ (thermal conductivity) = 3–6 W m$^{-1}$ K$^{-1}$ (ref. 47)) was conformally deposited by atomic layer deposition. In addition, for SHG polarimetry experiments a 100 nm thick Ag film was deposited on the top via electron beam evaporation, creating a heat sink (κ = 400 W m$^{-1}$ K$^{-1}$ (ref. 48)). Following these steps, the backside of the SiN$_x$ membrane is etched down to 50 nm to enable electron as well as incident laser beam transparency (Fig. 1a, inset).

**Optical measurements.** A femtosecond pulsed Ti: sapphire laser (Chameleon), tuned from 680 to 1,080 nm with $\sim 140$ fs pulse width and 80 MHz repletion rate, was utilized to excite SHG from individual GeTe nanowires via a home-built microscope equipped with a × 60, 0.7 NA objective (Nikon). It was incident through 50 nm thick SiN$_x$ membrane and focused with the spot size of $\sim 2$ μm. Its polarization was linearly controlled by a half-wave plate and chosen to be 0°, 60° and 120° with respect to NW's long axis. Under each excitation polarization (0°, 60° and 120°), the SHG signals were imaged by a cooled charge-coupled device and detected by a spectrometer (Acton) with a 300 groove mm$^{-1}$ 500 nm blaze grating with a charge-coupled device detector (Princeton instruments) with a spectral resolution of 0.1 nm. The polarization properties of SHG signal was studied by placing a polarizer in front of the detector[18].

**Electrical measurements.** Electrical pulses were applied using Keitheley 3401, and voltage-controlled d.c. *I–V* sweeps were performed using Keithley 2602 (*I–V* analyzer). Keithley 2700 was used as the data acquisition system (DAQ), and resistances were measured using Keithley 2602 via low bias ($-2$ mV to 2 mV) *I–V* sweeps.

**Electron diffraction simulation.** Electron diffraction patterns were simulated using the crystal-maker software (http://www.crystalmaker.com/) by super-imposing diffraction patterns of a bi-crystal of GeTe with the polarization vectors of each crystal (domain) at 71° and 109° to each other. The zone axis was set to another $\langle 111 \rangle$ direction, which is different from the polarization directions of each domain.

**Data availability.** The data that support the findings of this study are available from the authors upon reasonable request.

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

## Acknowledgements

This work was supported by the Office of Naval Research (grant N00014-16-1-2350), National Science Foundation (DMR-1505127 and 1210503), and Penn-MRSEC (NSF DMR-1120901). *In situ* electron microscopy experiments were performed at the Singh Center for Nanotechnology at the University of Pennsylvania. The authors are grateful for the insights provided by Prof Peter Davies during their discussions.

## Author contributions

P.N., Ra.A., M.R. and Ri.A. conceived the ideas and designed the experiments. Ri.A. lead the project. P.N., M.R. and G.L. performed the optical SHG experiments, and analysed the data. P.N. and Ra.A. carried out *in situ* TEM experiments, with assistance in electrical set-up from J.B. The in situ holder was designed and home built by ATCJ. Device fabrication for both TEM as well as SHG experiments were carried out by P.N. Results were discussed by all the authors. P.N. and Ri.A. co-wrote the manuscript.

## Additional information

**Competing interests:** The authors declare no competing financial interests.

