## [Peer Review File · Nature Communications]

Reviewers' comments:

Reviewer #1 (Remarks to the Author):

The authors have addressed adequately the issues I raised previously.

Reviewer #2 (Remarks to the Author):

Responses to the Round 1 Author Responses to Reviewer 2:

1) Ferroelectricity in GeTe – We might have to agree to disagree on this matter. The Reviewer looked at the data in the two papers suggested (Adv. Mater. 2016 and Appl. Phys. Lett. Mater. 2014) and such measurements – although, the Reviewer admits are widely and often inappropriately used to “confirm” ferroelectricity – can still be artifacts of a range of effects. One can produce such loops on a range of materials which are surely not ferroelectric. Look at the work on Vegard strains via chemical/ionic motion in materials for just one example. But perhaps this is beside the point at this time and we should not hold the current authors responsible for these prior works. Basically, I’m willing to let it go and I appreciate the change to “polar” throughout. This Reviewer also hopes the authors agree that the misuse of terms in the community – which is of course not their responsibility – is detrimental to the field in general and thus this is the motivation behind this stance.

2) Otherwise, I think the authors have honed in on my essential points – basically some seeming over promise of the language originally used, which is generally addressed in the responses.

Comments on new version of manuscript:

I have read the new version of text and have just one minor question: On page 12, the new text comparing to BiFeO₃ asks if the 180-degree boundaries are charged or uncharged? How could a 180-degree domain wall be charged? This is not clear to the reader.

Otherwise I believe the manuscript is improved and likely worthy of consideration for publication in Nature Commun.

Reviewer #3 (Remarks to the Author):

The authors have addressed most of the points I raised in the previous reports. The authors provide an improved manuscript with the quality of Nature Communications. Thus I recommend accepting the manuscript in Nature Communications after the authors make the following minor adjustments:

1. As reported by the authors, APB formation plays the main role to invert the polarization in germanium telluride, in which the mechanism is different from that of conventional ferroelectrics (such as domain nucleation and creep motion mechanisms in BaTiO₃ and BiFeO₃). Can the authors estimate the switching speed of the ferroelectricity driven by electrical pulses in germanium telluride?

2. Using electrical pulses to control the ferroelectricity is relatively new in metallic germanium telluride, however, it is definitely not a common way for conventional ferroelectric materials (such

as BaTiO₃, BiFeO₃, PZT and so on). Given that the switching speed, energy consumption of ferroelectrics are always the dominant consideration towards real applications, can the authors include some descriptions on the pros and cons of using heat-shock with current pulses for altering the polarization in ferroelectric materials (as compared with the electric-field driven switching in conventional ferroelectrics)?

Reviewer #1 (Remarks to the Author):

The authors have addressed adequately the issues I raised previously.

Reviewer #2 (Remarks to the Author):

Responses to the Round 1 Author Responses to Reviewer 2:

1. Ferroelectricity in GeTe – We might have to agree to disagree on this matter. The Reviewer looked at the data in the two papers suggested (Adv. Mater. 2016 and Appl. Phys. Lett. Mater. 2014) and such measurements – although, the Reviewer admits are widely and often inappropriately used to “confirm” ferroelectricity – can still be artifacts of a range of effects. One can produce such loops on a range of materials which are surely not ferroelectric. Look at the work on Vegard strains via chemical/ionic motion in materials for just one example. But perhaps this is beside the point at this time and we should not hold the current authors responsible for these prior works. Basically, I’m willing to let it go and I appreciate the change to “polar” throughout. This Reviewer also hopes the authors agree that the misuse of terms in the community – which is of course not their responsibility – is detrimental to the field in general and thus this is the motivation behind this stance.

Response: In order not to propagate half-truths, we had immediately accepted the referee’s earlier suggestion of calling GeTe polar. We are glad that the referee appreciates this. Furthermore, we share a similar criticism as that of the referee that Vegard strains originating from electromigration do contribute to P-V hysteresis in PFM, and that Adv. Mater.2016 and APL Mater. 2014 do not address these concerns. This discussion is beyond the scope of our work.

2) Otherwise, I think the authors have honed in on my essential points – basically some seeming over promise of the language originally used, which is generally addressed in the responses.

Response: Thanks.

Comments on new version of manuscript:

I have read the new version of text and have just one minor question: On page 12, the new text comparing to BiFeO3 asks if the 180-degree boundaries are charged or uncharged? How could a 180-degree domain wall be charged? This is not clear to the reader. Otherwise I believe the manuscript is improved and likely worthy of consideration for publication in Nature Commun.

Response: The 180-degree domain wall in GeTe contains a plane of Te-Te bonds (Fig. 4g), with head to head polarizations on either side of the domain wall. This results in some extra bound charge at the domain boundary itself. However, the presence of a sea of electrons could either completely screen these charges, or just partially (as predicted for another polar metal in Nature Communications, 7, #11211 (2016)). We now clarified this in the revised manuscript.

Reviewer #3 (Remarks to the Author):

The authors have addressed most of the points I raised in the previous reports. The authors provide an improved manuscript with the quality of Nature Communications. Thus I recommend accepting the manuscript in Nature Communications after the authors make the following minor adjustments:

1) As reported by the authors, APB formation plays the main role to invert the polarization in germanium telluride, in which the mechanism is different from that of conventional ferroelectrics (such as domain nucleation and creep motion mechanisms in BaTiO₃ and BiFeO₃). Can the authors estimate the switching speed of the ferroelectricity driven by electrical pulses in germanium telluride?

Response: A good estimate for the kinetics of ferroelastic relaxation after the formation of APB can be obtained from the speed of sound in GeTe (~1900 m/s). For a 2 μm long nanowire, this transformation should take place in a nanosecond. However, the existence of dislocations and other defects can act as pinning sites for domains, delaying the transformation kinetics. The formation of APB itself however could be a slower process, as it initially involves diffusion of Ge vacancies onto one Ge plane, before their eventual diffusionless collapse. Vacancy migration and mass motion is well-known in resistive memory devices such as Metal-TiO₂-metal, where mass diffusion kinetics are ultra non-linear with respect to the applied voltage (for e.g. see Nat. Comm. 6, #8610, 2015). This means that the time for switching or in our case APB formation, can be non-linearly decreased to few tens of nanoseconds just by increasing the external writing voltage. All in all, the entire switching process can be tuned externally to a few tens of nanoseconds. A brief discussion has been included in the revised manuscript.

2) Using electrical pulses to control the ferroelectricity is relatively new in metallic germanium telluride, however, it is definitely not a common way for conventional ferroelectric materials (such as BaTiO₃, BiFeO₃, PZT and so on). Given that the switching speed, energy consumption of ferroelectrics are always the dominant consideration towards real applications, can the authors include some descriptions on the pros and cons of using heat-shock with current pulses for altering the polarization in ferroelectric materials (as compared with the electric-field driven switching in conventional ferroelectrics)?

Response: As already adequately discussed, the biggest advantage of heat-shock assisted polar domain switching is that this can be applied on polar metals and leaky-ferroelectrics where electric field cannot be efficiently utilized. Since this process involves extracting work out of heat (heat shock), and if and when applied in devices, it should be a lossy process and more power-consuming than conventional ferroelectrics switched via electric field (and no heat). However, it must be realized that this is the only way of switching polar metals experimentally *discovered thus far*. Switching speeds although (see the response above), should be comparable to conventional ferroelectric switching.

REVIEWERS' COMMENTS:

Reviewer #2 (Remarks to the Author):

I'm am happy to accept the manuscript at this time.

Reviewer #3 (Remarks to the Author):

The authors have replied detailedly for the reviewers' questions; also, they have improved their revised manuscript. Thus, I suggest its publication at Nature communications.